# Asynchronous suppression of visual cortex during absence seizures in *stargazer* mice

Jochen Meyer [1], Atul Maheshwari[1], Jeffrey Noebels[1] & Stelios Smirnakis[2,3]

Absence epilepsy is a common childhood disorder featuring frequent cortical spike-wave seizures with a loss of awareness and behavior. Using the calcium indicator GCaMP6 with in vivo 2-photon cellular microscopy and simultaneous electrocorticography, we examined the collective activity profiles of individual neurons and surrounding neuropil across all layers in V1 during spike-wave seizure activity over prolonged periods in *stargazer* mice. We show that most (~80%) neurons in all cortical layers reduce their activity during seizures, whereas a smaller pool activates or remains neutral. Unexpectedly, ictal participation of identified single-unit activity is not fixed, but fluctuates on a flexible time scale from seizure to seizure. Pairwise correlation analysis of calcium activity reveals a surprising lack of synchrony among neurons and neuropil patches in all layers during seizures. Our results demonstrate asynchronous suppression of visual cortex during absence seizures, with potential implications for understanding cortical network function during EEG states of reduced awareness.

[1] Department of Neurology, Baylor College of Medicine, One Baylor Plaza, Houston, TX, USA. [2] Department of Neurology, Brigham and Women's Hospital, Harvard Medical School, Boston, MA, USA. [3] Jamaica Plain Campus, VA Boston Healthcare System, Boston, MA, USA. These authors contributed equally: Jochen Meyer, Atul Maheshwari. Correspondence and requests for materials should be addressed to J.M. (email: jfmeyer@bcm.edu) or to S.S. (email: smsmirnakis@bwh.harvard.edu)

Absence epilepsy interrupts normal cortical processing, producing reversible episodes of altered consciousness. Each seizure begins without warning, replacing planned motor movements with speech arrest and a vacant stare lasting only a few seconds, followed by sudden and complete recovery of awareness and intentional behavior. The events provide unique functional insight into the coupling of human perception and volition, and the cellular basis of this seizure type is steadily emerging from the study of genetic mouse models. The *stargazer* model, one of over 20 monogenic mouse mutants with this phenotype[1], displays frequent, recurrent spike-wave seizures with behavioral arrest that are sensitive to blockade by ethosuximide[2]. Loss of the transmembrane AMPA receptor regulatory protein (TARP) subunit Cacng2 in *stargazer* mice leads to mistrafficking of dendritic AMPA receptors in fast-spiking interneurons in the neocortex[3] and thalamus[4,5] and to remodeling of firing properties in thalamocortical circuitry that favor abnormal oscillations[6]. This loss of inhibition, particularly feedforward inhibition, is implicated in the pathophysiology of most monogenic models of absence epilepsy[1,7]. However, prior work in humans and animal models of absence epilepsy reveal inconsistent evidence of where and how cortical activity is modulated during seizures. While several studies in rats showed no activity changes in visual cortex[8], or in somatosensory and motor cortex[9], functional magnetic resonance imaging (fMRI) studies in humans have demonstrated increased ictal blood oxygen level-dependent (BOLD) activity in the occipital cortex[10], or biphasic activation and deactivation of large scale networks, including visual cortex[11,12]. Here, we combine simultaneous electrocorticography and 2-photon microscopy to overcome both the poor spatiotemporal resolution of fMRI and the limited number of neurons that can be simultaneously recorded with patch-clamp electrophysiology. Surprisingly, we find a suppression of both activity and synchrony in all layers of primary visual cortex of *stargazer* mice during electrographic spike-wave discharges.

## Results

**Visual cortex is suppressed during absence seizures**. Functional activity within large populations of GCaMP6-labeled neuron somata ($n = 29–132$ per field) and patches of neuropil ($n = 4–18$ per field) were readily visualized over prolonged periods in awake mice across all layers (Fig. 1a; Supplementary Figure 1). Neurons with exceptionally low activity (~12.6% of all labeled neurons) were excluded from the analysis (see Methods). A total of 22 data sets from 15 mice were analyzed (Supplementary Table 1), generating a total of 1366 distinct neurons and 222 patches of neuropil. For each data set, the average activity of both neurons and neuropil in the ictal state ($\Delta F/F$, mean ± s.e.m. of neurons = 5.3 ± 1.7%; neuropil = 0.8 ± 0.2%) was significantly lower than the activity in the interictal state (neurons = 8.1 ± 2.2%; neuropil = 3.6 ± 0.5%, paired $t$-test, $p < 0.001$, Fig. 2a, b). Calcium activity of individual neurons was then aligned to cortical electroencephalogram (EEG) seizure onset or offset (Fig. 1b), and distributions of calcium activity were compared between the interictal and ictal states using the Wilcoxon Rank Sum Test (see methods). Three patterns of ictal activity emerged (Fig. 1a–d). In all cortical layers, most neurons had lower activity during ictal periods (mean ± s.e.m., 82.3% ± 2.4%, "ictal-low" neurons in Fig. 2c), while significantly fewer neurons showed either higher activity during ictal periods ("ictal-high", 5.7% ± 0.9%, one-way ANOVA, $p < 0.005$, Fig. 2c) or no significant change ("neutral", 12.1% ± 1.9%, $p < 0.005$, Fig. 2c). Neuronal firing rate was also estimated using a validated deconvolution algorithm[13,14] (Supplementary Figure 2), revealing similar dominance of ictal-low over ictal-high neurons (Supplementary Figure 3). Deconvolution

of calcium activity was validated with simultaneous cell-attached recordings and calcium imaging, showing excellent correlation between action potentials (APs) and visualized calcium transients (Supplementary Figure 2). Patch-clamp recordings from a subset of animals ($n = 16$ putative pyramidal cells in 8 animals) further verified the predominant ictal-low firing patterns of layer 2/3 (L2/3) *stargazer* neurons (Ictal-low in 14/16, ictal-high in 2/16 ($p = 0.009$, Wilcoxon paired rank-sum), Supplementary Figure 4). In addition, all neuropil patches, containing overlapping dendritic and axonal processes from many nearby neurons, were significantly ictal-low (Fig. 2b). These findings demonstrate that the visual cortex is in a largely "suppressed state" during absence seizure events.

**Hypoactivity starts several seconds before seizure onset**. We next evaluated the time course of engagement for both neurons and neuropil by comparing calcium activity ($\Delta F/F$) in half-second windows to a baseline ($\Delta F/F$) generated from circularly shuffling the seizure time points (Kolmogorov–Smirnov test, $p < 0.05$ after correction for multiple comparisons). Starting 10 s before and up to 5 s after seizure onset, an average of 33.1% ± 6.1% of neurons displayed significant change in activity ($\Delta(\Delta F/F)$) relative to seizure onset in at least one interval of two consecutive 0.5-s bins, thereby indicating significant "participation" in seizure events. Using a similar 15-s window spanning 5 s prior to until 10 s following seizure offset, significant activity changes were found on average in 41.1% ± 6.7% of neurons. Layer 2 (L2) ictal-low neurons displayed gradually reduced activity starting about 7 s prior to seizure onset and then remained significantly below mean activity between 1 s prior to onset and 1 s after offset (Fig. 2d). Neuropil activity displayed a gradual reduction earlier than neurons, dropping significantly below baseline 4 s prior to onset, but it returned to baseline around the same time as ictal-low neurons (Fig. 2e). Neurons and neuropil in deeper layers showed similar reductions with layers 4 and 5 (L4 and L5) dropping below baseline within several seconds of seizure onset, and layer 6 (L6) at 2–3 s prior to onset (Supplementary Figure 5). In contrast to ictal-low neurons, only 21.7% of ictal-high and 31.6% of neutral neurons had significant activity changes in at least 2 consecutive 0.5-s bins relative to baseline, with no consistent pattern relative to seizure onset and offset.

**Loose temporal participation of neurons with absence seizures**. Given the variable participation of neurons relative to seizure onset and offset, we next asked to what degree each neuron remains within its overall classified activity profile over time, versus flipping to the opposite category. To do this, we compared ictal to interictal activity ($\Delta F/F$) inside a sliding 7-min window in data sets from all layers, which advanced from seizure to seizure over the course of the recording. The choice of a 7-min window was empirically found to balance the power necessary for statistical analysis with temporal resolution. Figure 3a plots $p$-values for significance for each window in a typical ictal-high neuron, displaying 1 epoch of consecutive 7-min windows which were in the predicted ictal-high state (red, nine consecutive seizures). In this example, there was also 1 ictal-low epoch which stretched over four consecutive seizures (blue). As expected, a significantly greater percentage of time was spent in the predicted classification compared to the opposite classification (21.5 ± 3.3% vs 3.2 ± 1.1% over all neurons, $p < 0.0001$, paired $t$-test, Fig. 3b). Ictal-low neurons, however, had a significantly greater likelihood of remaining in the predicted classification (33.3 ± 4.4% vs 7.9 ± 3.1%, $p < 0.001$) and a significantly lower likelihood of being transiently classified in

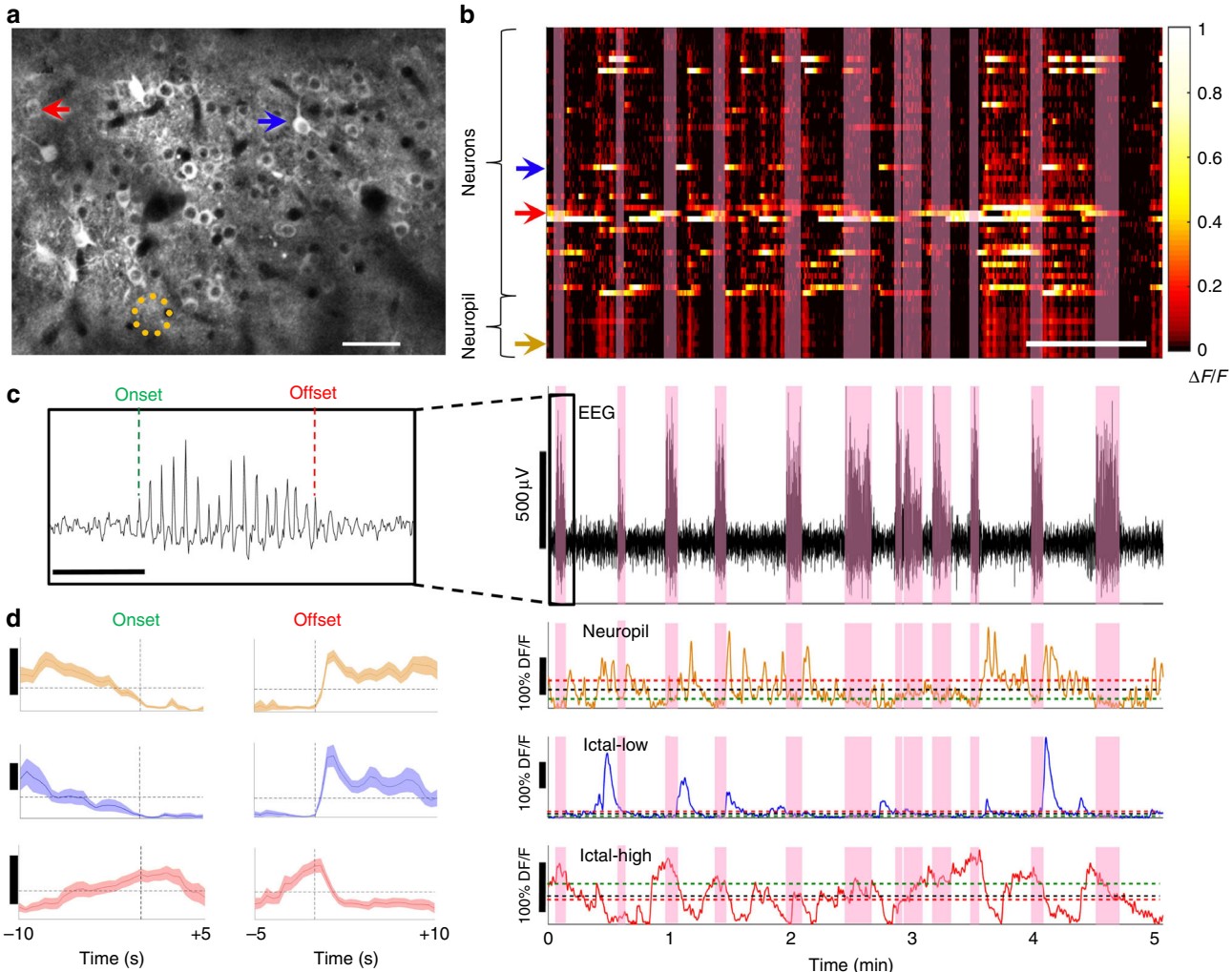

**Fig. 1** In vivo 2-photon microscopy and EEG in visual cortex of *stargazer* mice. **a** Typical field of view for in vivo imaging; GCaMP6-filled neurons in L2/3 of visual cortex (red arrow—neuron with high activity during seizures (ictal-high); blue arrow—neuron with low activity during seizures (ictal-low); orange dotted circle = neuropil patch. Scale bar = 50 μm. **b** Top: Calcium activity for each neuron/neuropil patch over time with seizure epochs highlighted in pink (horizontal bar = 1 min). Bottom: concomitant EEG, and traces of calcium activity from the depicted neuropil, ictal-low neuron, and ictal-high neuron shown in **a**. Mean ictal activity (green dashed line), mean interictal activity (red dashed line), and overall mean activity (black dashed line) are plotted. Inset **c** shows definition of seizure onset and offset at first and last spike of the seizure, respectively (bar = 1 s). **d** Average activity (mean ± s.e.m., vertical bar = 20% $\Delta F/F$) of an exemplary neuropil patch (top), ictal-low neuron (middle), and ictal-high neuron (bottom) are shown, time-locked to the seizure onset (left) and offset (right)

the opposite classification (0.4 ± 0.1% vs 6.5 ± 2.6%, $p = 0.012$) than ictal-high neurons overall across all layers (2-way ANOVA with post-hoc Bonferroni correction, Fig. 3b). To evaluate the transient nature of seizure coupling over a longer time interval, the participation of defined L2/3 neurons was evaluated for the same cortical window at multiple time points over the course of 8–12 days. Figure 3c–f presents an example. In total, for three chronically recorded animals with three recordings at least 24 h apart, a single neuron had a 44.0 ± 1.2% probability of changing its classification to any other class between any two time points. 60.6 ± 2.9% of all neurons (ictal-low, ictal-high or neutral) changed at least once over the total time recorded. More than half of ictal-low neurons (57.9 ± 2.9%) maintained a consistent classification throughout the three time points whereas ictal-high neurons were never found to be consistently ictal-high. Altogether, these data show that neocortical neuron activity can be loosely coupled to seizures over both short- and long-range time intervals, with ictal-low neurons having greater consistency than ictal-high neurons.

**Pairwise synchrony is reduced during absence seizures**. We next examined synchrony between and within nearby neurons and surrounding neuropil during interictal and ictal states using pairwise Pearson correlation coefficients. Since there is a mathematical reduction in correlation that occurs when average pairwise firing rates decrease, we corrected for baseline activity as previously described (Methods)[15,16]. The average correlation between pairs of neurons was significantly reduced in the ictal (0.09 ± 0.01) compared to the interictal state (0.16 ± 0.02), and neuropil to neuropil patch correlations were similarly reduced during seizures (ictal: 0.62 ± 0.03, interictal: 0.81 ± 0.02, Fig. 4a–c, e, Wilcoxon matched-pairs signed rank test, $p < 0.005$). As expected, intra-neuropil pairwise correlation coefficient strength was significantly greater than intra-neuron pairwise correlation in both the interictal and ictal states ($p < 0.001$). When subdivided into specific groups, only ictal-low neurons were significantly less correlated in the ictal state, whereas ictal-high and neutral neurons had no significant change in correlation (Fig. 4d). This pairwise reduction of correlation in ictal-low neurons had no

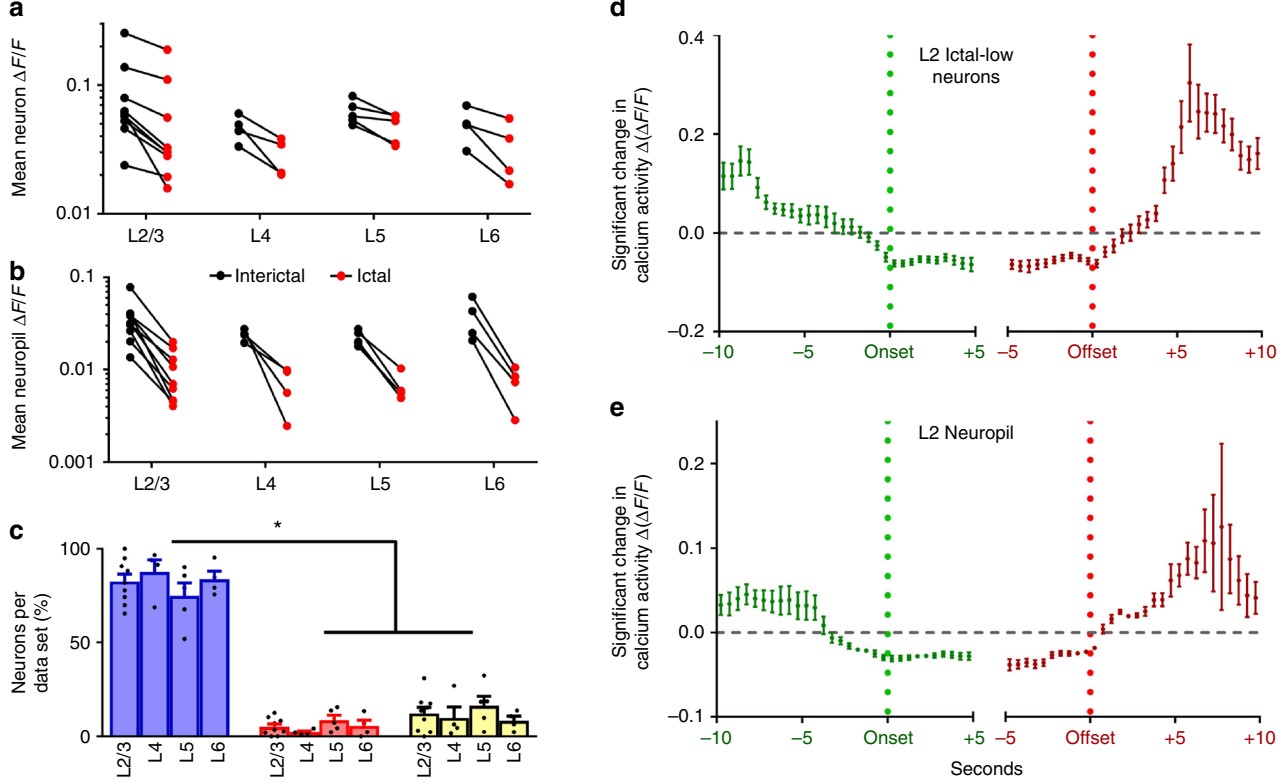

**Fig. 2** Activity changes in the ictal state. **a** Average calcium activity ($\Delta F/F$) was significantly reduced in the ictal state compared to the interictal state for neurons across all layers. Line segments connect interictal (black disk) and ictal (red disk) mean activity within a field of view. Each comparison was statistically significant based on the Wilcoxon matched-pairs signed rank test ($p < 0.01$). **b** Similar findings in neuropil ($p < 0.001$ for all comparisons). **c** Significantly greater percentage of ictal-low compared to ictal-high or neutral neurons was seen across layers ($p < 0.0001$ across data sets; 2-way ANOVA with Tukey's multiple comparisons. **d** Temporal profile of significant change in ictal-low neuron activity compared to baseline, where $\Delta(\Delta F/F)$ represents deviation from the average $\Delta F/F$. Note that mean $\Delta(\Delta F/F)$ (±s.e.m.) starts falling about 7 s before seizure onset, returning to baseline following seizure offset (n/bin = 9–155 neurons). **e** Temporal profile of change in neuropil activity shows that neuropil activity starts falling gradually more than 5 s prior to seizure onset, with a similar return to baseline after seizure offset (n/bin = 2–53 neuropil patches)

relationship with the reduction in pairwise geometric mean calcium activity ($r^2 = 0.003$, Supplementary Figure 6), further validating that the decrease in synchrony occurs independently of the reduced activity seen during the ictal state. Analysis with deconvolved traces (Supplementary Figure 6B, 7) yielded similar results. In addition, since locomotion itself has been associated with greater activity in the visual cortex[17], it was important to demonstrate that our observations are not the indirect result of different locomotion rates in the interictal versus the absence (ictal) state. Two mice were analyzed after digitally subtracting image frames occurring during locomotion. As expected, only 0–1% of ictal frames coincided with locomotion, compared to 9.4% and 17.6% of frames in the interictal state of each data set. Importantly, removal of locomotion-associated frames did not affect our basic observation that significantly lower neuronal activity and lower inter-neuronal synchrony occur in the ictal state (Supplementary Figure 8).

Given the reduced synchrony between neurons during seizures seen at the temporal resolution of two-photon imaging (200 ms bins), we next evaluated to what degree single-unit spikes correlate with seizure spikes on the EEG, at a temporal resolution of 20 ms bins. Pooled distributions of the timing of all APs recorded in L2/3 ($n = 13,131$ APs from 16 neurons) relative to the EEG spike times (taken at time 0) revealed only two (12.5%) neurons demonstrating a peak clustered around the time of EEG spikes, while the remaining 14 (87.5%) neurons had no significant temporal relation to the EEG spike times (D'Agostino & Pearson normality test, $p < 0.05$, Supplementary Figure 9). The mean

likelihood of a neuron spiking within ± 20 ms around an EEG spike was only $17.9 \pm 3.9\%$ (mean ± s.e.m). Given the overall low neuron-EEG spike synchrony, these findings do not support elevated synchrony of visual cortical neurons during spike-wave seizures at a higher temporal resolution.

**PV+ and Som+ interneurons are suppressed during seizures.** To test if ictal classification may to some degree depend on cell type, we drove tdTomato expression in *Dlx5/6-Cre* (Dlx+) and *Somatostatin-Cre* (Som+) *stargazer* mice. Crossing *Parvalbumin-Cre* (PV+) with *stargazer* mice was limited by syntenic expression on chromosome 15[3]. We therefore employed the CLARITY technique[18] with post-hoc immunostaining to identify parvalbumin-expressing interneurons (Fig. 5). Out of 88 identified GABAergic interneurons across layers 2–5, 84 (95.5%) were ictal-low (55 of 57 Dlx+, 12 of 13 Som+, and 17 of 18 PV+ interneurons), while the remainder were neutral neurons. No ictal-high neurons were found among all GABAergic neurons. Therefore, our data indicate that ictal classification of medial ganglionic eminence (MGE)-derived interneurons does not differ significantly from neuronal classification overall.

## Discussion

Our results in the visual cortex differ from a previous electro-physiological description of neuronal firing in the somatosensory cortex[9]. In genetically inbred rats with absence epilepsy, intracel-lular recordings from a limited number ($n = 8$–17) of single cortical

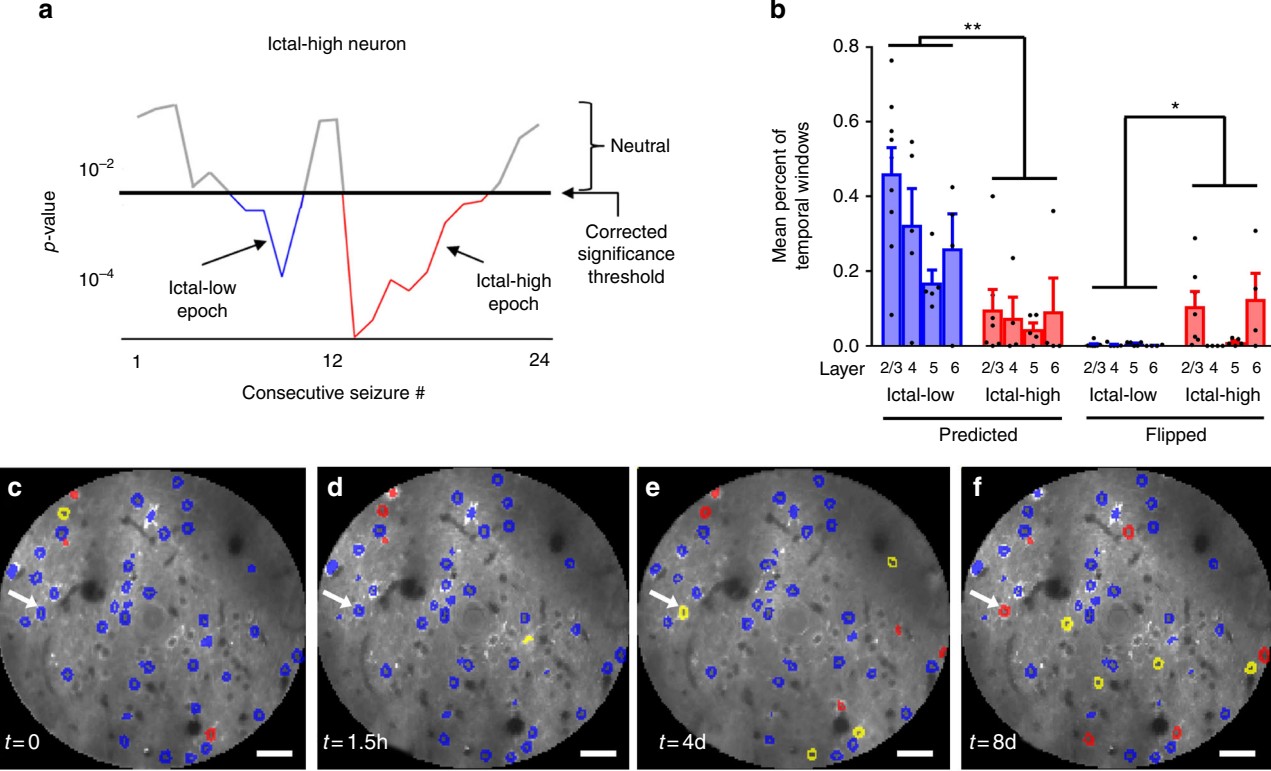

**Fig. 3** Temporal participation of neurons over minutes to days. **a** With a 7-minute moving window, a neuron with an overall classification of "ictal-high" had 1 significant ictal-high segment (red), periods of no significant difference (gray), and it also flipped to have 1 brief ictal-low epoch (blue). **b** Temporal windows were more often in the predicted than the flipped classification. In addition, ictal-low neurons had a smaller proportion of flipped classifications than ictal-high neurons (*$p = 0.028$, **$p < 0.001$). Bar plots: mean (±s.e.m.). **c** Distribution of ictal-high and ictal-low neurons in one data set, **d** 90 min later, **e** after 4 days, and **f** after 8 days in the same chronically imaged window. Note, for example, that one neuron starts as ictal-low but then becomes neutral after 4 days and ictal-high after 8 days (arrow). Blue = ictal-low, red = ictal-high, yellow = neutral, uncolored = quiet (neurons with exceptionally low average activity in at least one of the four time points; Methods). Horizontal scale bars in **c–f**: 35 μm

neurons showed no significant change in firing rates during absence seizures[9,19] with unit firing synchronized with the EEG spike discharge. Here, in contrast, simultaneous imaging of a large sample of primary visual cortex (V1) neurons revealed that mean neuronal activity ($\Delta F/F$) is significantly reduced during the ictal state across all cortical layers. This is reflected in the fact that the large majority of neurons (81.5%) and neuropil are ictal-low, while ~15% of neurons show no change in activity and only ~5% increase their activity during seizures. Calcium signaling within neuropil has been shown to be more consistent with EEG activity[20], and so our findings are consistent with the hypothesis that the electrographic spike-and-wave seizures of absence epilepsy are largely "inhibitory" seizures[21]. This is the first evidence at a cellular level that visual cortex is in a significantly hypoactive state during absence seizures, and is consistent with previous findings that cortical neurons have limited overall pathological excitation[22,23] when they engage in absence compared to convulsive seizures[24–27].

Previous fMRI studies of spike-wave epilepsy in humans have shown clear regional differences in activation, including signal increases during ictal activity in visual cortex[10] or biphasic modulations in brain-wide networks, including the visual cortex[12]. One fMRI study in a rat model of absence epilepsy, however, showed no BOLD activity changes in visual cortex during seizures[8]. The variability among these reports and the difference from our results may be in part due to frequency dependent neurovascular coupling over a range of spike-wave discharge frequencies (3–11/s) in these models[1,28], the lower spatiotemporal resolution of the BOLD signal, and the fact that the BOLD signal may also reflect inhibitory activity. Our results also revealed that

the cellular activity was loosely coupled to the seizures, with neurons dynamically changing their activity with respect to seizures over the course of minutes to days. This lack of a hardwired cellular network correlate with a stereotyped EEG pattern is also consistent with recent findings in focal epilepsy in both mouse[27] and human[29] recordings.

Surprisingly, we also found a significant reduction in pairwise correlation strength between nearby cortical neurons during seizure events within the visual cortex, most pronounced in ictal-low neurons and surrounding neuropil, which was significant even after correcting for the reduced level of activity and reduced time spent in the ictal state. These findings are in direct contrast to spike-wave complexes associated with focal onset seizures, which show significantly increased pairwise synchrony in the ictal state in vitro[30] and in humans in vivo[25]. It is interesting to consider what this means in terms of information encoding[31,32], and whether it could explain why absence seizures neither preclude the cortical transmission of evoked potentials[33] nor have a postictal state. This issue will require further study in the context of stimulus presentation, taking also into account the effect that the accompanying subthreshold membrane voltage modulation is likely to have for information transmission across different areas.

Finally, it is of interest to speculate about the potential difference between ictal-low and ictal-high neurons. In a previous study of single units during focal seizures in patients with epilepsy, suppressed neurons were typically regular-spiking, presumably excitatory neurons, while increased activity was found in fast-spiking, presumably inhibitory neurons[34]. However, in our sampling of interneurons derived from the MGE (Dlx+, PV+,

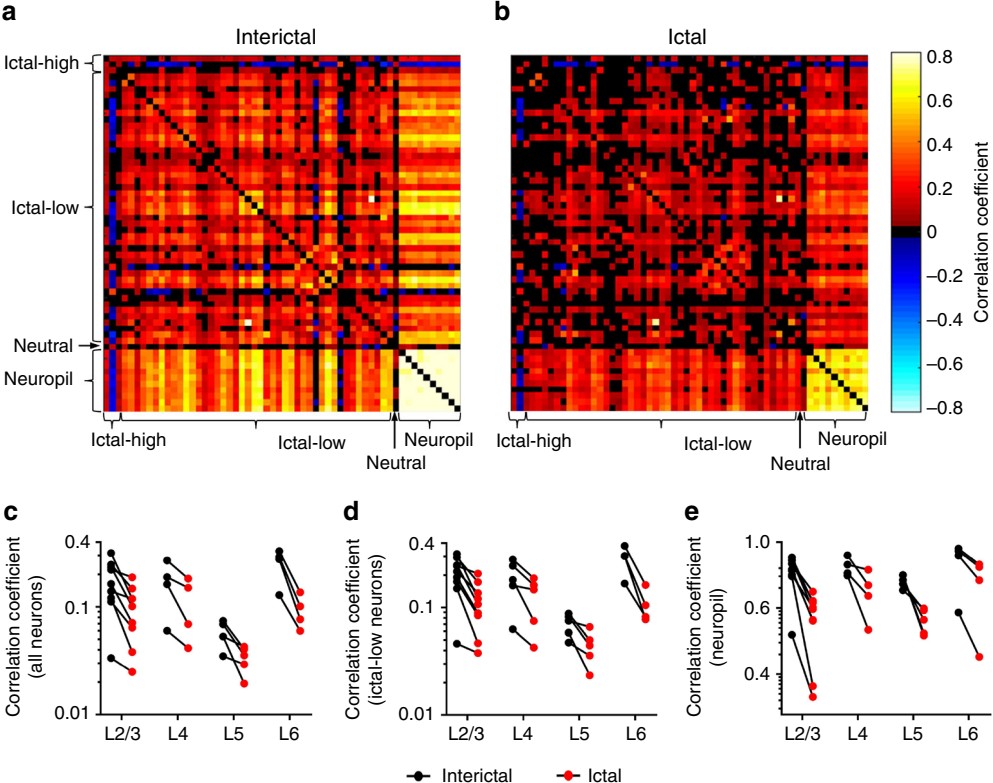

**Fig. 4** Pairwise synchrony is reduced in the ictal state. Pairwise correlation coefficients from a representative example of L2/3 neurons and neuropil in the **a** interictal and **b** ictal states. Across all data sets in all layers, there was a significant reduction in synchrony in the ictal state for **c** all neurons, **d** ictal-low neurons, and **e** neuropil ($p < 0.005$, Wilcoxon matched-pairs signed rank test)

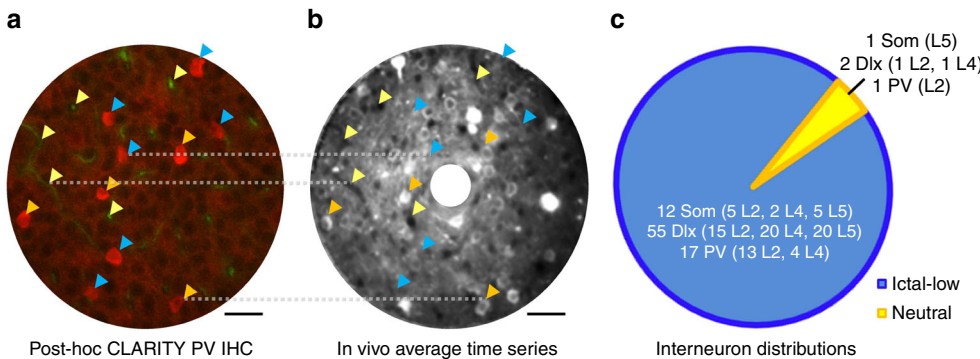

**Fig. 5** Interneurons identified in vivo and ex vivo. **a** Unperfused, clarified brain tissue allowed blood vessels to auto-fluoresce in the green channel (yellow arrowheads), while PV+ interneurons were immunostained and are shown in red (blue and orange arrowheads). Horizontal scale bar = 30 μm. **b** Co-registration with blood vessels (yellow arrowheads labeling both in-plane and perpendicular vessels) from an in vivo average of a time series revealed PV+ interneurons that were GCaMP6+ (orange arrowheads) versus GCaMP6- (blue). Horizontal scale bar = 30 μm. **c** Combined with interneurons identified in vivo due to expression of tdTomato under *Somatostatin-Cre* (Som) or *Dlx-5/6-Cre* (Dlx) drivers, the majority of identified interneurons were ictal-low in nature

SOM+), the vast majority of these neurons had reduced activity in the ictal state. PV+ neurons in particular consistently showed reduced activity as predicted due to an AMPA receptor trafficking deficit previously described in these neurons[3], which is consistent with a prior study showing that optogenetic inactivation of PV+ neurons in visual cortex leads to reduced pairwise correlation among cortical neurons[35]. Another source of suppression may be from a distinct class of interneurons including neurogliaform or vasoactive intestinal peptide (VIP)-expressing neurons. Since our findings were based on calcium imaging reflecting AP firing, the apparent widespread synchrony of the EEG may arise from synchronized subthreshold oscillations, which could be further

dissected with simultaneous multi-cell patching and voltage-sensitive dyes. In line with the specific behavioral arrest associated with generalized spike-and-wave seizures, the asynchronously suppressed activity we observed in visual cortex provides a new framework for understanding cortical network malfunction leading to loss of awareness during absence ictal states, and for further exploring the cellular underpinnings of conscious experience.

## Methods

**Surgical implantation and chronic in vivo calcium imaging**. All procedures were carried out according to a protocol approved by the IACUC at Baylor College of

Medicine. A minimum sample size of eight mice was estimated using a power level set at 0.8 to detect an effect of at least 25% for each paired variable tested, with an estimated 4% variance between paired differences. AAV-1 GCaMP6M (11 animals) or GCaMP6S (1 animal) virus with a neuron-specific promoter (*AAV1.Syn. GCaMP6m.WPRE.SV40*, U Penn) was injected (100 nL) at 2 different depths (250 and 550 μm) using a Nanoject II automatic nanoliter injector through a burr-hole over the right V1 (2.5 mm lateral and 1.5 mm anterior to lambda) of 6-week-old *stargazer* mice of either sex under isoflurane anesthesia. The deeper injection was located ~300 μm further medial. This made it possible to image as deep as 760 μm below the dura due to the lack of overlying Gcamp6-expressing structures contributing out-of-focus fluorescence[36]. For some of the L4 through L6 imaging experiments, we used two different lines expressing Cre and tdTomato selectively in L4 (*Scnn1a-Cre x Ai9*) or L6 (*Ntsr1-Cre x Ai9*), and crossed these mice to the *stargazer* line. F1 offspring positive for Cre, Ai9 and heterozygous for the *stargazin* mutation were bred together, and a subset of their offspring were homozygous *stargazer* mutants expressing both Cre and tdTomato. 1 mm long, flat Ag/Ag-Cl electrodes were placed epidurally over the ipsilateral somatosensory cortex, 2 mm anterior to the middle of the craniotomy, and over the contralateral visual cortex; and a titanium headpost was permanently attached with dental cement. A reference electrode was implanted over the contralateral cerebellar hemisphere. A 3 mm diameter glass cranial window was placed over the injected area and imaging was performed following a 4–6-week recovery period to allow sufficient cellular expression of the calcium indicator. Mice were imaged awake, head-posted in a holding frame and allowed to run freely on a circular treadmill. Raw calcium image series were acquired with a Prairie Ultima IV 2-photon microscope using a ×25 objective, 1.1 NA, or a ×16 objective, 0.8 NA, at 890 nm under spiral or raster galvo scanning mode (10–20 Hz frame rate). In nine recordings from individual animals, we targeted L2/3 (mean depth below pia: 163 μm, range 100–240 μm). We also imaged five fields of view (FOVs, from four animals) in L4 (mean depth below pia: 383 μm, range 360–395 μm), five FOVs (from three animals) in L5 (mean depth below pia: 570 μm, range 510–640 μm), and four FOVs (from three animals) in L6 (mean depth below pia: 730 μm, range 660–760 μm). Cells in L4 were identified either by using *stargazer × Scnn1a-Cre/Ai9* line which expresses tdTomato in L4 pyramidal neurons, or by targeting the typical depth of L4 (330–480 μm)[37], ensuring that cell bodies appeared relatively smaller than in L2/3. L5 was identified either by using the *stargazer × Scnn1a-Cre/Ai9* line and focusing right below L4, or by targeting the typical depth of L5 (480–680 μm)[37], where pyramidal cell bodies are significantly larger than in L4. L6 was identified either by using the *stargazer × Ntsr1-Cre/Ai9* line which expresses tdTomato exclusively in L6 pyramidal neurons, or by targeting the typical depth of L6 (680–900 μm)[37], where the largest pyramidal cell bodies are smaller than in L5 (on average 214 μm² in L5[38] versus on average 133 μm² in L6, based on our own measurements in two *stargazer* x *Ntsr1-Cre* mice). Laser output power under the objective was kept below 50 mW, corresponding to ~20% of power levels shown to induce lasting histological damage in awake mice[39]. In L5 and L6 FOVs, laser power was higher (up to 150 mW), but by visual inspection for signs of bleaching, cell swelling, and other damage, we are confident that we did not cause physical harm to the neurons. This is because the power delivered per unit volume remains low as the total volume over which the power is distributed increases.

During imaging sessions, the EEG signal was sampled at 2 or 5 kHz with filtering cut-offs set at 1 Hz and 250 Hz. Seizures were detected by visual inspection by an experienced user (AM) in MATLAB (EEGLab) according to specific criteria (regular spike-wave burst structure, spike amplitude 1.5× baseline, spike frequency of 5–9 Hz, and a minimum duration of 0.5 s). The peaks of the first and last EEG spike were considered as the seizure onset and offset, respectively (Fig. 1c).

In five animals, we recorded locomotion during the recording sessions using an angular velocity sensor. In addition, we monitored animal behavior by recording wide-field video of the mouse using an IR camera (Thorlabs DCC3240N) at 20 fps.

Patch-clamp recordings were performed in 8 *stargazer* and 2 control animals (*n* = 16 cells, 7 whole-cell, 9 cell-attached, plus 2 whole-cell (control)) for validation of the calcium-signal, as well for confirmation of the differences in ictal vs. interictal activity. We used borosilicate microelectrodes (4–7 MΩ) for current-clamp recordings and a previously published recipe for the current-clamp solution[15], (in mM): 105 K-gluconate, 30 KCl, 10 HEPES, 10 phosphocreatine, 4 ATP-Mg, and 0.3 GTP. Data were acquired at 10 kHz and analyzed using custom-written MATLAB routines. Briefly, spike times were converted into firing rates and smoothed with a 100 ms sliding window. This window length was found empirically in patched neurons that expressed GCaMP6 by determining the highest correlation coefficient between their deconvolved traces and their actual smoothed spike rates using different smoothing window sizes. For further analysis, the resulting firing rate traces were subjected to the same algorithms used for analysis of the calcium activity traces described below.

**Analysis of calcium imaging**. The raw calcium fluorescence sequences were run through a custom *x/y* motion correction algorithm (MATLAB Inc.), based on a previously published method[40]. Briefly, motion parameters were estimated in the red channel in recordings from animals that had a subclass of interneurons labeled with tdTomato, otherwise the green channel was used, and motion-corrected movies were visually inspected for errors and discarded if necessary. All image frames were registered to the average of the first five image frames using a sub-pixel

registration method. Then, the correction offsets were applied to the green channel to reconstruct motion-corrected movies. Animals were resting on the treadmill during and between seizures. Since this seizure type is characterized by behavioral arrest, ictal movement was minimal. On average, only 7% of all frames per recording had some motion artifact that was corrected, with an average excursion of the frames exhibiting motion artifact of 1.5 μm. Gradual movement of <5 μm was reliably compensated for by the motion correction algorithm. Seizures with movement >5 μm occurring within 2 s before and after seizure onset or offset were excluded from further analysis[41]. To ensure that there was no significant contamination of cellular calcium activity by neuropil activity (especially in the z-axis), we tested a subset of recordings and confirmed that subtraction of neuropil activity from nearby cellular calcium signals did not affect the outcome of our analysis.

Regions of interest (ROIs: neuronal cell bodies and neuropil patches) were identified manually with ImageJ or semi-automatically using a custom MATLAB routine under supervision. The MATLAB routine was based on a previously published algorithm[42]: A seed point for a local ROI was placed manually with a circular ring to cover the cell body. Around the center of the cell, the algorithm then computes an intensity profile in polar coordinates. The final selection of the pixels for the ROI matches the maximal intensity along the polar profile. Neuropil patches were chosen to be larger than cell bodies because dendrites of local neurons labeled with GCaMP6, in contrast to other calcium indicators such as OGB, can have very high-amplitude fluorescence, which would dominate the signal of a neuropil patch if it consisted of relatively few pixels. Because we sought to analyze the signal comprised of contributions from a large number of axons and dendrites, comparable to LFP, neuropil patches for each data set were, on average, 682 ± 96 μm² (mean ± s.e.m.). Neuropil patches were always further than 5 μm beyond the outline of ROIs defined as neurons.

Raw calcium traces for each ROI were created using the mean of all pixel intensities inside an ROI, and then high-pass filtered (0.1 Hz) and normalized by their individual baselines to ΔF/F values. To differentiate calcium transients from baseline fluctuations, the baseline (F) at time point t was defined as the mean of the bottom 10% of all data points within $t \pm 20$ s, similar to prior work[41]. Noise was identified and removed from the ΔF/F signal by plotting a histogram of the negative data points of each ROI's ΔF/F trace, fitting it with a half-Gaussian function, identifying the 0.5-SD distance below zero, and removing all data points below the peak of the Gaussian +0.5-SD. We used simultaneous patch-clamp and calcium imaging data to calibrate this procedure. We also performed a similar analysis after deconvolving the calcium traces and found similar results. Firing rates were extrapolated from the ΔF/F traces using a modified version of 2 different deconvolution methods[13,14], yielding consistent results. The method used to deconvolve the data we present in the text was based on a previously published method[13] that uses (1) an iterative smoothing process to remove local low-amplitude peaks representing noise without distorting the ΔF/F signal stemming from calcium fluctuations, and (2) inverse filtering of the smoothed traces with an exponential kernel. The other method was used for further validation of the results and infers an approximation of the most likely spike train underlying the given fluorescence trace using an iterative expectation maximization algorithm.

We empirically determined a minimum level of activity for an ROI below which it was not possible to determine whether the ROI had statistically significant firing events due to lack of data points, by calculating the sum of noise-corrected ΔF/F signal per minute for each ROI. ROIs with mean activity below this threshold (sum (ΔF/F)/min = 6 for GCaMP6M, sum(ΔF/F)/min = 22.5 for GCaMP6S) were deemed "quiet" (12.6 ± 1.6% of all neurons) and excluded from further analysis[42]. The remaining ROIs were then classified as ictal-high, ictal-low or neutral by comparing all activity during the ictal state with all activity in the interictal state using the Wilcoxon rank-sum test (MATLAB) with Bonferroni correction for multiple comparisons. Comparison between and within group data was performed in Prism 5 (version 5.0d, GraphPad, CA, USA).

**Temporal analysis of calcium imaging**. To minimize cross-contamination due to the relatively slow calcium signal dynamics, seizures (and adjacent inter-seizure intervals) that were shorter than 1.5 s, as well as seizures that were <6 s apart, were also excluded from analysis.

To determine the temporal evolution of each ROI's classification around seizure onset and offset, calcium activity was aligned to seizure onset and offset separately, and null distributions of calcium activity were created by circularly shuffling the seizure time points 2000 times while keeping the seizure length and calcium activity data constant (MATLAB). Thirty bins of activity at 0.5-s intervals were created spanning from 5 s prior to until 5 s after seizure onset as well as from 5 s prior to until 10 s after seizure offset. The difference between the activity for each bin relative to seizure onset or seizure offset was compared against the corresponding bin of the null distribution to determine if the activity (ΔF/F) in a given ROI was significantly increased or decreased at that particular temporal window (MATLAB, Kolmogorov–Smirnov test, significance set at $p < 0.0017$ (0.05/30) for correction of multiple comparisons. To be conservative, we undertook the further step of requiring that at least two neighboring 0.5-s bins reached significance.

In order to characterize the behavior of each ROI on a finer temporal scale, we applied the same classification analysis to sliding windows of 7 min of activity along the duration of each recording. Each window was then advanced serially by one

seizure. Seizure start and end times were then evaluated within any given 7-min window, ictal time periods and interictal time periods were concatenated within each window and binned in 2-s intervals, and the Wilcoxon rank-sum test (MATLAB) was performed to compute significance between the ictal and interictal states.

**Correlation analysis.** Pairwise Pearson correlation coefficients were calculated (MATLAB) separately for the interictal versus the ictal states as follows: the amplitude of each ROI's ictal activity was normalized by the average amplitude of the ROI's interictal activity. In order to correct for the mathematical reduced likelihood of correlation created by reduced activity, for each pair of neurons, we circularly shuffled one neuron's concatenated ictal and interictal time points separately 2000 times and computed null distributions of correlation coefficient strengths. The mean of the null distributions was then subtracted from the unshuffled original coefficients independently for the interictal and ictal states as previously described[16]. Alternatively, we randomly removed calcium-events from interictal (if interictal activity was higher) or ictal (if ictal activity was higher) time periods, until ictal and interictal average $\Delta F/F$ rates were <0.1% apart for each cell. Analyzing concatenated epochs versus averaging correlation coefficients from individual epochs yielded similar results, indicating that the concatenation process did not artificially affect correlations (data not shown).

**Analysis of patch-clamp electrophysiological data.** To determine whether patched neurons were more active during versus between seizures, we identified AP times in voltage traces from cell-attached or whole-cell recordings (custom MATLAB routine) and calculated the ictal and interictal firing rates. APs were identified by searching for any time points $t$ in the voltage trace that met the following criteria:

i.  $V_m(t+0.25\ ms) > V_m(t) + x$,
ii.  $\text{Mean}(V_m(t\text{-pre}_1\ \text{to}\ t\text{-pre}_2)) < \text{mean}(V_m(t+p_1\ \text{to}\ t+p_2)) - \alpha \times x$, and
iii.  $\text{Mean}(V_m(t+\text{post}_1\ \text{to}\ t+\text{post}_2)) < \text{mean}(V_m(t\text{-}\ \text{pre}_1\ \text{to}\ t\text{-}\ \text{pre}_2)) + \beta \times x$,

where the threshold $x$ was 38 mV, $\text{pre}_1$ was 1.2 ms, $\text{pre}_2$ was 0.3 ms, $p_1$ was 0.25 ms, $p_2$ was 0.35 ms, $\text{post}_1$ was 2.5 ms, $\text{post}_2$ was 2.9 ms, $\alpha$ was 0.5 and $\beta$ was 0.8 for whole-cell recordings, and $x$ was 1.9 mV, $\text{pre}_1$ was 0.8 ms, $\text{pre}_2$ was 0.25 ms, $p_1$ was 0.15 ms, $p_2$ was 0.24 ms, $\text{post}_1$ was 1.8 ms, $\text{post}_2$ was 2.1 ms, $\alpha$ was 0.55 and $\beta$ was 0.45 for cell-attached recordings. In each recording, we identified the seizure spikes and used the spike times as zero-points to generate the peri-spike time histograms (from $-100$ to $+100$ ms) that indicate to what degree APs were time-locked to the spikes of a seizure.

**Code availability.** Custom-written MATLAB routines will be made available per request to the authors.

**Data availability.** All data supporting the findings will be made available by request to the authors.

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

## Acknowledgements

We acknowledge funding from: NINDS R21 NS088457 (J.M. and S.S.), NINDS K08 NS096029-01 (A.M.); and NINDS NS29709 (J.N.).

## Author contributions

J.M. carried out the imaging and electrophysiology experiments, analyzed data, and edited the manuscript, A.M. carried out the CLARITY experiments, analyzed data, and wrote the manuscript. S.S. and J.N. conceived and supervised the project, and edited the manuscript.

## Additional information

**Competing interests:** The authors declare no competing interests.

