## [Peer Review File · Nature Communications]

Reviewers' comments:

Reviewer #1 (Remarks to the Author):

In this manuscript, Meyer et al. use 2-photon calcium imaging to investigate L2/3 visual cortical activity during absence seizures. They find that despite large amplitude EEG activity, most neurons in L2/3 are suppressed during ictal events (seizures). They then find that single neurons are not consistent across ictal events, but can change their level of enhancement/suppression. Finally, they argue that cortical neurons are surprisingly desynchronized during ictal events. These results have implications for our understanding of cortical processing during absence seizures.

I will preface this review by clarifying that while I have expertise in visual cortical processing and 2-photon microscopy, I have only passing experience with absence epilepsy, and defer to the other reviewers' comments on the contribution of this paper to the epilepsy field.

The experiments were generally carried out in a sensible way, but I have some concerns about their analyses that make me question some of their findings.

Major concerns:

1) In several cases the authors use circular permutation tests to test the significance of ictal-high or ictal-low activity. However, given the slow time course of GCaMP6, I don't believe that adjacent permutations can be considered independent samples. Assuming that these permutations are independent leads to underestimation of the null distribution variance and overestimation of the single cell significance. This would explain how single cells within a seven minute window could somehow reach significance levels of $p \ll 10^{-50}$ (a number which evokes astronomical scales). I would recommend that the authors check with a statistician to ensure the validity of their permutation analyses.

2) The correlation analysis has a major issue with decreased activity during the ictal periods. The authors claim that their permutation tests account for this, but I do not see how it resolves the issue of low activity rates leading to near-zero between-cell correlations. A better way to account for the low activity levels during ictal events would be to use deconvolution to convert calcium to spikes, then randomly subtract spikes from the interictal periods until the mean spike rate is the same between ictal and interictal periods, then calculate pairwise correlation coefficients on the rate-matched data.

3) As mentioned above, I am not an expert in the epilepsy field, but these results seem to contradict earlier results showing that cortical neurons are highly synchronized during absence seizures (e.g., Seidenbecher et al., *J Eur Neurosci*, 1998; Chipaux et al., *PLoS One*, 2013), is there some explanation for this discrepancy? Perhaps a difference in animal model?

Minor points:

1) I believe the correct protein name is GCaMP6, not GCamp6.

2) Figure 1B, bottom: no numbers on time axis.

3) Fig 2C: This plot is confusing, it would be preferable to have the entire curve shown in grey with the significant segments highlighted with color.

4) Layer 4 imaging results only show highly processed data. Supp Fig. 7 should show an example field (similar to Fig 1A) and a matrix of neural responses (similar to Fig 1B).

General comments:

1) The authors did not leverage one of the major advantages of genetically encoded calcium indicators. This study would have benefitted considerably from imaging activity in excitatory neurons and inhibitory neurons separately (e.g., are the occasional ictal-high cells all inhibitory?). It's probably too late to add to the study now, but they should strongly consider this if they continue the line of experiments.

Reviewer #2 (Remarks to the Author):

In this manuscript, the authors perform elegant in vivo calcium imaging experiments to examine how the superficial layer-2/3 neurons signal before, during, and after spike-and-wave seizures in the stargazer model of absence epilepsy. Combining calcium imaging recordings with simultaneous EEG recordings positioned them to characterize how a relatively large population of neurons behave during these SWDs. The majority (~80%) of the neurons in layer-2/3 in visual cortex show reduced calcium signaling (a measure of activity) during an SWD. In a subset of mice, the authors also observed an average reduction of activity in neurons and neuropil in Layer 4 neurons. The authors interpret these data to state that the majority of neurons in the layer-2/3 visual cortex are asynchronously hypoactive during seizures.

The study is elegant and the quality of the results is very good. However, I have concerns regarding the experimental design, data interpretation and novelty.

Strengths:

1. The major strength of this paper is the ability to image and infer activity of "many" neurons at one time, as opposed to previous studies in which recordings from single cortical neurons during absence seizures reveal that there was not a significant change in firing rate.
2. The methods are excellent and the statistical analysis and description of tests taken to ensure that studies are sufficiently powered really help to solidify the data collected and presented. The authors describe clearly in the methods how those seizures that were studied had to be significantly long (> 1.5 seconds).
3. The authors performed in vivo whole cell patching to further verify the activity patterns from the in vivo calcium imaging studies, which really helps to solidify the interpretation of their data.
4. The authors performed impressive chronic imaging recordings from the same neurons, 3 sessions spread out over 8-12 days, to see if neuron activity clusters with seizures. This approach allowed them to reliably interpret that layer-2/3 neurons in visual cortex are loosely coupled with SWDs in their absence epilepsy model.

Moreover, this work potentially has interesting ramifications for differences between absence and convulsive seizures. In the SWDs, since the majority of the cortical neurons are hypoactive, it could explain why SWDs do not develop into GTCS, since cortical neurons appear less prone to runaway pathological hyperexcitation.

Weaknesses/Areas for improvement.

1. The novelty of this study seems to be the hypoactivity in layers 2/3 that the authors find surprising and in contrast with the previous reports that found robust rhythmic activation of cortical neurons during SWDs in a rat model of absence epilepsy (Polack et al., 2007 J Neurosci). However, such comparison is not appropriate. Are the results in the superficial cortical layers truly "in contrast to prior expectations"? The first paper cited from the Charpier Lab (Polack et al., 2007 J. Neurosci) shows intracellular recordings from layer 2/3 neurons in GAERS during absence seizures (Fig. 2 of their paper); however, these recordings were done in S1 and not V1. The second paper from the Charpier lab (Polack and Charpier, 2006 J Physiol) examines cortical neuron firing patterns during high-voltage rhythmic spikes (which they argue represent absence-like seizure activity) in the Long-Evans Rats. In this paper the cortical neurons were recorded in the orofacial motor cortex and although the mean firing rate was not significantly different during the

HVRS, the standard deviation of the ISI decreased dramatically during the HVRS suggesting the cortical neurons were firing more rhythmically.

The point here is none of these studies were looking at patching *in vivo* in V1 and in layer 2/3 during SWDs. This point gets at the novelty of the finding.

Notably, the rationale for looking at visual cortex over somatosensory cortex is not clear and needs to be clarified. If the authors wanted to compare their results with previous findings, why did they choose a different cortex?

The somatosensory cortex may have been more tractable for SWDs? Comparing layer 2/3 cells in visual cortex to layer 5 cells in the somatosensory cortex is not straightforward. If the authors had recorded layer 2/3 cells in the somatosensory cortex, they might have found that these cells are active similar to Polack et al 2007.

2. The main finding in Polack et al., 2007 was that deep S1 neurons initiate SWDs. In this study however, the authors are excluding the first 2 seconds of a seizure event in order to avoid contamination by calcium signal in preceding time period, meaning these results do not get at any dynamics of SWD initiation. Can the authors better support the “unexpected” nature of their finding?

It seems more likely that no one has done this before and not that the authors are finding new results with this approach.

3. How might changes in layer 2/3 neurons in visual cortex impact thalamocortical circuitry, which is known to be important for SWDs? In lieu of citing primarily fMRI data in the introduction, it would be useful to cite some patching studies as well that provide a more mechanistic insight (such as Tan et al., 2007, PNAS).

4. The paper would benefit from a more thorough discussion of the neuropil results. They do not define this clearly, there is no clear rationale for measuring their activity separately (i.e. no relevant background information), and little discussion about the implication of the neuropil results.

5. Can the authors elaborate on the “new framework” (i.e. how is asynchronous activity compatible with high-amplitude EEG events)?

6. Can the authors provide an explanation for why they did not choose to use the GCaMP6f variant? In their discussion of “temporal analysis of calcium imaging” (pg. 17), authors note that they exclude the first 2 seconds of all ictal and interictal segments in order to prevent contamination by the preceding time period, based on their results that the de-noised calcium signal reliably returns to baseline within 2 seconds after an action potential is fired. This is a potential concern, given that the average seizure duration was 4 seconds (reported in Supp. Table 1). Authors note that they exclude seizures that are shorter than 1.5 seconds (presumably after excluding the first 2 seconds, so these seizures would have had to be at least 3.5 seconds). They do not address the possibility that this exclusion criterion is biasing their analysis/observation towards longer events which may be qualitatively different than shorter events. Would the authors still face this issue if they had used the vast GCaMP6 variant?

According to Chen et al., 2013, Figure 3, GCaMP6s and GCaMP6m have slower kinetics compared to GCaMP6f. Decay time of 6m is ~2-3 x greater than that of 6f, decay time of 6s is ~5-6x greater than that of 6f.)

6. Layer 2/3 is not a homogeneous cell population. The authors should put this study into a more circuit-level context. I understand that they are trying to get at the cellular substrates, but a brief discussion of the visual cortical microcircuitry would be helpful in order to aid in speculations about the role/identity/function of these identified “ictal-high” and “ictal-low” neurons. For instance, what are L2/3 inputs/outputs?

7. What the effects are of the 21% of neurons with low activity that were not included during analysis? In theory, these were neurons that did not "respond" to different SWD state (before, during or after). How does the exclusion of this data affect the robustness of the finding that the "majority" of neurons reduce their activity during SWDs, and how does this relate to previous studies? The use of Cre-lines would be helpful to know if these non-responders were all a similar cell type. One report from Tan et al. 2007, PNAS shows that in a specific model of childhood absence epilepsy in which a point mutation occurs in the GABAR $\gamma 2$ (R43Q) subunit, that layer 2/3 pyramidal neurons have reduced GABAR currents, while nRT and TC neurons had no reduction in GABAR-mediated currents. This could suggest that some of the ictal-low neurons were in fact GABAergic neurons.

The authors used Syn promoter in this study which is not specific to a cell type. It would have been interesting to know which of the neurons are excitatory versus GABAergic, perhaps based on the expression of the GCamp6M or GCamp6S that would be restricted to excitatory or inhibitory cells using different Cre lines (see Khoshkoo et al., Cell, 2017).

Minor points:

1. Although they find asynchrony during ictal periods in the superficial L2/3 neurons, this suggests asynchrony within the layer. They also find similar results when looking at L4 neurons in 2 mice. It would be interesting to see whether there is a/synchrony across cortical layers, using the available data.

2. In the example traces shown in Supplementary Figure 2, are the authors showing activity during a seizure? Or is this normal activity? They should show seizure activity to show how the GCaMP6 signal can follow it, similar to Khoshkoo et al., 2017.

Reviewer #3 (Remarks to the Author):

In this paper, the authors use a mice model of absence epilepsy, with the technique of 2-photon GCaMP6 Ca imaging, to monitor the activity in cerebral cortex during absence seizures. They find that most neurons in superficial layers reduce their activity during the seizure, and do not display synchronized activity. This is a nice study, possibly important for the understanding of seizure activity, but only if a couple points can be fixed.

1. I do not understand the statement that superficial layers form "distinct" networks, since the authors did not record in deep layers, how do they know? It could be that the authors record from a brain area that is completely silent, including deep layers. Alternatively, it could be that only superficial layers are suppressed. Since it was shown previously that these experimental models of absence seizure have a focus, and since it is known that the focus is surrounded by a perifocal zone (the "penumbra"), where the activity is very much depressed (as in Schwatz & Bonhoeffer, 2001, for example). So it may be that the authors just record from this perifocal zone, and thus it would not be surprising to find a suppressed firing during the seizure.

The only way to further shed light into this would be to provide some recordings of deep layers in the same preparation. I understand that it may not be possible to image in deep layers, so I suggest to the authors to include a couple of unit recordings (simultaneous multi-unit recordings, from superficial and deep layers, would suffice). Only then, they can claim that the superficial network is distinct or disconnected.

If this is provided, it would greatly increase the value and impact of the study.

2. Since layer 4 also seems suppressed during seizures, wouldn't this argue that indeed the whole area might be suppressed? This also calls for having recordings in deep layers.

3. Similar findings were reported from single-unit recordings during focal seizures in human temporal cortex (Dehghani et al., Sci Reports 6: 23176, 2016, see the last figures in this paper), where it was also found that many neurons in Layer 2-3 are suppressed during the seizure. However, in that study, it was found that most of the suppressed neurons are RS (regular-spiking) cells, while those who increased firing are mostly FS (fast-spiking) cells, which were presumably inhibitory. This study should certainly be compared to the present results, and there should be a discussion of whether the ictal "low" and "high" cells found here could also be RS and FS cells, in which case it would completely agree with the phenomenon observed by Dehghani et al.

This would further stress the resemblance between absence seizures and focal epilepsy, which is another argument supporting a focal origin for absence epilepsy. I think this would be worth being discussed as well.

Reviewer #1:

1) In several cases the authors use circular permutation tests to test the significance of ictal-high or ictal-low activity. However, given the slow time course of GCaMP6, I don't believe that adjacent permutations can be considered independent samples. Assuming that these permutations are independent leads to underestimation of the null distribution variance and overestimation of the single cell significance. This would explain how single cells within a seven minute window could somehow reach significance levels of $p \ll 10^{-50}$ (a number which evokes astronomical scales). I would recommend that the authors check with a statistician to ensure the validity of their permutation analyses.

We agree with the reviewer's comment regarding independence, but this is the very reason we selected to construct the null distribution via circular permutations. Circular permutations preserve the "internal" statistical properties of the calcium time series (i.e. the shape of the calcium responses) while destroying the temporal relationship of the time series with seizures or other cell's time series. This is in fact the statistically recommended way to construct a null distribution that is appropriate for testing the significance of temporal relationships across different time series (for example, whether cells fire more during a seizure or whether the cross correlation of the calcium signal across pairs of cells is significant). In other words, circular permutations retain the part of the variance that is due to the shape of the calcium signal and do not lead to an overestimation of significance (as opposed to constructing a null distribution by random – not circular – reshuffling of the time series, which would).

This concern is further mitigated by our validation with deconvolved data where the decay portion of the calcium signal has effectively been removed prior to reshuffling.

We also note that permutations were not used when testing whether a cell was ictal-low or ictal-high (when we could directly compare the ictal versus the interictal periods), but only when determining the significance of seizure participation around seizure start and end times. We further ensured that accepted permutations were more than two seconds away from the original data to make sure the calcium tail was excluded.

Finally, the seven-minute moving window analysis is also solely based on the Wilcoxon rank sum test without permutations, where activity was aligned to seizure start and end times. We now make this clearer in the methods section on pages 19-20.

2) The correlation analysis has a major issue with decreased activity during the ictal periods. The authors claim that their permutation tests account for this, but I do not see how it resolves the issue of low activity rates leading to near-zero between-cell correlations. A better way to account for the low activity levels during ictal events would be to use deconvolution to convert calcium to spikes, then randomly subtract spikes from the interictal periods until the mean spike rate is the same between ictal and interictal periods, then calculate pairwise correlation coefficients on the rate-matched data.

We have re-run the correlation analysis using these recommended methods, and the results are unchanged, with overall reduced synchrony during the ictal state ($p=0.0005$, Wilcoxon matched-pairs signed rank test). These results have been added to Supplementary Figure 5.

3) As mentioned above, I am not an expert in the epilepsy field, but these results seem to contradict earlier results showing that cortical neurons are highly synchronized during absence seizures (e.g.,

Seidenbecher et al., J Eur Neurosci, 1998; Chipaux et al., PLoS One, 2013), is there some explanation for this discrepancy? Perhaps a difference in animal model?

Seidenbecher et al (J Eur Neurosci, 1998) has shown synchrony between EEG discharges and simultaneous recordings in single units in the thalamus and somatosensory cortex. Chipaux et al., PLoS One, 2013 has also shown synchrony between single neurons in the somatosensory cortex and the EEG. However, neither has shown synchrony distributions between a large population of individual cortical neurons occupying the same lamina before, during, and after these seizures. In addition, we are looking primarily at the visual cortex, which now appears to have different levels of activity and synchrony during seizures than the somatosensory cortex. We have provided additional data (Supplemental Figure 9), which also highlights the relative lack of synchrony between visual cortex neurons and the EEG.

Minor points:

1) I believe the correct protein name is GCaMP6, not GCamp6.

This has been corrected.

2) Figure 1B, bottom: no numbers on time axis.

This has been corrected.

3) Fig 2C: This plot is confusing, it would be preferable to have the entire curve shown in grey with the significant segments highlighted with color.

We agree that this figure is confusing and does not add substantively to the manuscript. This has been removed.

4) Layer 4 imaging results only show highly processed data. Supp Fig. 7 should show an example field (similar to Fig 1A) and a matrix of neural responses (similar to Fig 1B).

This has been corrected and can be seen in Supplementary Figure 1.

General comments:

1) The authors did not leverage one of the major advantages of genetically encoded calcium indicators. This study would have benefitted considerably from imaging activity in excitatory neurons and inhibitory neurons separately (e.g., are the occasional ictal-high cells all inhibitory?). It's probably too late to add to the study now, but they should strongly consider this if they continue the line of experiments.

We agree with the comment of the reviewer, and clearly the use of Cre lines is of potentially great benefit. Since *stargazin* and *parvalbumin* are only 0.01 cM away from each other on chromosome 15, we were not able to breed our *stargazer* line with the *Parvalbumin-Cre* line. However, we were able to breed onto both *Dlx 5/6-Cre* and *Somatostatin-Cre* lines to identify subsets of interneurons. We also used CLARITY and post-hoc immunohistochemistry to identify parvalbumin-positive neurons in a subset of recordings. The data from those studies summarized are on page 9 (Figure 5).

Reviewer #2 (Remarks to the Author):

Weaknesses/Areas for improvement.

1. The novelty of this study seems to be the hypoactivity in layers 2/3 that the authors find surprising and in contrast with the previous reports that found robust rhythmic activation of cortical neurons during SWDs in a rat model of absence epilepsy (Polack et al., 2007 J Neurosci). However, such comparison is not appropriate. Are the results in the superficial cortical layers truly “in contrast to prior expectations”? The first paper cited from the Charpier Lab (Polack et al., 2007 J. Neurosci) shows intracellular recordings from layer 2/3 neurons in GAERS during absence seizures (Fig. 2 of their paper); however, these recordings were done in S1 and not V1. The second paper from the Charpier lab (Polack and Charpier, 2006 J Physiol) examines cortical neuron firing patterns during high-voltage rhythmic spikes (which they argue represent absence-like seizure activity) in the Long-Evans Rats. In this paper the cortical neurons were recorded in the orofacial motor cortex and although the mean firing rate was not significantly different during the HVRS, the standard deviation of the ISI decreased dramatically during the HVRS suggesting the cortical neurons were firing more rhythmically. The point here is none of these studies were looking at patching in vivo in V1 and in layer 2/3 during SWDs. This point gets at the novelty of the finding.

Notably, the rationale for looking at visual cortex over somatosensory cortex is not clear and needs to be clarified. If the authors wanted to compare their results with previous findings, why did they choose a different cortex? The somatosensory cortex may have been more tractable for SWDs? Comparing layer 2/3 cells in visual cortex to layer 5 cells in the somatosensory cortex is not straightforward. If the authors had recorded layer 2/3 cells in the somatosensory cortex, they might have found that these cells are active similar to Polack et al 2007.

We agree. The reason we chose visual cortex was because our particular interest was in understanding how sensory neocortical networks behave during the absence seizure, not necessarily to probe the origin of the seizure focus as determined in rat models of absence. In support of this we had 1) observed that spike-wave seizures appeared the same in visual cortex as in somatosensory or any other cortical area by recording with a surface electrode from exactly above the location that was imaged in one of the mice; 2) we have previously shown stargazin to be present in PV+ interneurons in visual cortex of mouse (Maheshwari et al, Frontiers Cell Neurosci 2013) and 3) inhibition/excitation, connectivity and physiology using two-photon methods have been studied extensively in visual cortex. We now emphasize our choice of this sensory cortex in the title and limit our conclusions accordingly. We do plan on performing similar recordings in somatosensory cortex for the reasons the reviewer stated. We have added language to the discussion (page 10) to make it clear that the findings we see in the visual cortex are potentially different from the somatosensory cortex. We think that this is a very important issue for further investigation, and we have begun to collect data from other areas including somatosensory cortex, but it is too preliminary to make conclusions and we feel that it is outside the scope of this manuscript.

2. The main finding in Polack et al., 2007 was that deep S1 neurons initiate SWDs. In this study however, the authors are excluding the first 2 seconds of a seizure event in order to avoid contamination by calcium signal in preceding time period, meaning these results do not get at any dynamics of SWD initiation.

Our aim in this paper is not to obtain a causal proof of SWD initiation. However, we do find that the activity changes seen in cortical neurons and neuropil precede the onset of the first observed EEG spike

(see Fig. 2D-E and Supplementary Figure 5) by several seconds. The only place we originally excluded the 2 second analysis was for this participation analysis. In any event, to prevent any confusion, we have changed the analysis to avoid omitting any time points.

Can the authors better support the “unexpected” nature of their finding? It seems more likely that no one has done this before and not that the authors are finding new results with this approach.

We agree with the reviewer that these results may not be unexpected, and we have taken out this word. It is also true that no one has previously approached absence epilepsy in this way, and this approach yielded the following new findings (please see also discussion, page 10-11, for more detail): (1) suppression of activity in neocortex and (2) reduced synchrony between neurons during spike-wave seizures.

3. How might changes in layer 2/3 neurons in visual cortex impact thalamocortical circuitry, which is known to be important for SWDs? In lieu of citing primarily fMRI data in the introduction, it would be useful to cite some patching studies as well that provide a more mechanistic insight (such as Tan et al., 2007, PNAS).

We thank the reviewer and have now cited this study in our introduction (page 2). In that study, Tan et al showed reduced IPSCs onto pyramidal cells. This is similar to the proposed pathophysiology in *stargazer* mice, which also are expected to have reduced inhibition due to deficient trafficking of AMPA receptors to cortical PV+ interneuron dendrites. Defective inhibition on the pyramidal neurons alone in these two models would not lead to suppression of activity in visual cortex; therefore, we suspect there are other potential mechanisms for causing reduced activity, including contributions from various interneuron subtypes (see discussion, page 12).

4. The paper would benefit from a more thorough discussion of the neuropil results. They do not define this clearly, there is no clear rationale for measuring their activity separately (i.e. no relevant background information), and little discussion about the implication of the neuropil results.

We thank the reviewer for this suggestion. We have expanded this portion of the discussion on page 10, with the rationale that neuropil may better represent the activity of inputs arriving in the image layer (Kerr et al, 2005, PNAS), and correspond better to EEG/LFP activity.

5. Can the authors elaborate on the “new framework” (i.e. how is asynchronous activity compatible with high-amplitude EEG events)?

Since the EEG reflects subthreshold activity that is not captured by the calcium signal, the high-amplitude spike-wave seizures may largely arise from synchronous subthreshold oscillations. These can be seen in our patch-clamp recordings (Supplementary Fig. 8). However, the spiking of most neurons is not temporally locked with these oscillations, with only $17.9 \pm 3.9\%$ of spikes from patched neurons firing within ± 20 msec of an EEG spike (Supplementary Fig. 8, results on page 9). This low neuron-EEG spike correlation is consistent with ictal decorrelation of spiking activity as well as decreased firing of most neurons in the ictal state.

6. Can the authors provide an explanation for why they did not choose to use the GCaMP6f variant? In their discussion of “temporal analysis of calcium imaging” (pg. 17), authors note that they exclude the first 2 seconds of all ictal and interictal segments in order to prevent contamination by the preceding time period, based on their results that the de-noised calcium signal reliably returns to baseline within 2 seconds after an action potential is fired. This is a potential concern, given that the average seizure duration was 4 seconds (reported in Supp. Table 1). Authors note that they exclude seizures that are shorter than 1.5 seconds (presumably after excluding the first 2 seconds, so these seizures would have had to be at least 3.5 seconds). They do not address the possibility that this exclusion criterion is biasing their analysis/observation towards longer events, which may be qualitatively different than shorter events. Would the authors still face this issue if they had used the fast GCaMP6 variant? According to Chen et al., 2013, Figure 3, GCaMP6s and GCaMP6m have slower kinetics compared to GCaMP6f. Decay time of 6m is ~2-3 x greater than that of 6f, decay time of 6s is ~5-6x greater than that of 6f.)

We appreciate the reviewer’s thoughtful comments, but we felt the use of Gcamp6f had little additional value to this work because (1) it does not provide significantly better temporal resolution than the deconvolved traces derived from Gcamp6 M or S, which we have used to validate the activity and correlation analyses as seen in Supplementary Figures 2, 3 and 6; (2) It has a lower signal to noise ratio than M and S (see figure to the right from Chen et al, 2013) which would make it more challenging to delineate clear $\Delta F/F$ signal, especially in deeper layers; and (3) experiments that we have performed with OGB (similar kinetics to GCaMP6f) have similar results (data not shown, but can be provided as needed). Regarding the exclusion of data within 2 seconds of prior epochs, we have repeated our analysis without exclusion and reached the same conclusion.

6. Layer 2/3 is not a homogeneous cell population. The authors should put this study into a more circuit-level context. I understand that they are trying to get at the cellular substrates, but a brief discussion of the visual cortical microcircuitry would be helpful in order to aid in speculations about the role/identity/function of these identified “ictal-high” and “ictal-low” neurons. For instance, what are L2/3 inputs/outputs?

We now have evidence that the findings in Layer 2/3 are similar across layers. In wild-type animals the inputs to L2/3 are largely from L4 and outputs are largely to other cortical regions, but it is not known if these canonical circuits are perturbed alongside the development of epilepsy. In this setting, we find it difficult to speculate on the role ictal-high and ictal-low neurons play in the interictal versus ictal state, but we now include some circuit-level discussion on pages 10-11. We specifically found that Parvalbumin+ and Somatostatin+ neurons are largely suppressed, so the identity of ictal-high neurons will need further elucidation. This is particularly challenging given the relatively sporadic participation of ictal-high neurons on both small and large time scales.

7. What the effects are of the 21% of neurons with low activity that were not included during analysis? In theory, these were neurons then that did not “respond” to different SWD state (before, during or after). How does the exclusion of this data affect the robustness of the finding that the “majority” of neurons reduce their activity during SWDs, and how does this relate to previous studies?

After lowering the threshold for noise based on more careful calibration with simultaneous patch clamp recordings in cells with GCaMP6m expression, the proportion of “quiet” neurons reduced to ~12.6%. Previous work also shows silent or unresponsive neurons that were excluded from further analysis (e.g.

the 2013 GCaMP6 paper by Chen, now referenced in our methods section, p. 20), so we believe that our findings remain valid.

The use of Cre-lines would be helpful to know if these non-responders were all a similar cell type. One report from Tan et al. 2007, PNAS shows that in a specific model of childhood absence epilepsy in which a point mutation occurs in the GABAR $\gamma 2$ (R43Q) subunit, that layer 2/3 pyramidal neurons have reduced GABAR currents, while nRT and TC neurons had no reduction in GABAR-mediated currents. This could suggest that some of the ictal-low neurons were in fact GABAergic neurons. The authors used Syn promoter in this study, which is not specific to a cell type. It would have been interesting to know which of the neurons are excitatory versus GABAergic, perhaps based on the expression of the GCamp6M or GCamp6S that would be restricted to excitatory or inhibitory cells using different Cre lines (see Khoshkoo et al., Cell, 2017).

We have now added SST-Cre and Dlx 5/6-Cre lines and found no difference from overall neuronal activity (Figure 5C). We were unsuccessful in breeding PV-Cre lines with our *stargazer* line because of syntenic expression on chromosome 15 (only 0.01 cM apart). However, we were able to successfully identify PV-expressing neurons using post-hoc CLARITY (Figure 5A-B).

Minor points:

1. Although they find asynchrony during ictal periods in the superficial L2/3 neurons, this suggests asynchrony within the layer. They also find similar results when looking at L4 neurons in 2 mice. It would be interesting to see whether there is a/synchrony across cortical layers, using the available data.

We now have datasets from all layers and show asynchronous suppression throughout the visual cortex. Using the available data, we are not able to compare correlations between layers, but this is an area of interest, which we plan to pursue but outside the scope of this current work.

2. In the example traces shown in Supplementary Figure 2, are the authors showing activity during a seizure? Or is this normal activity? They should show seizure activity to show how the GCaMP6 signal can follow it, similar to Khoshkoo et al., 2017.

The example in Supplementary Figure 2 (now Supplemental Figure 3) was showing interictal activity, and demonstrates that calcium signal deconvolution can accurately predict spiking activity. We have now added an example of an **ictal high** neuron with a GCaMP signal that is loosely coupled to seizures in Supplemental **Figure 2A**. Figure 1D also shows a typical example of how two individual neurons, one ictal high (red) and one ictal low (blue), engage to the onset/offset of seizures.

Reviewer #3 (Remarks to the Author):

1. I do not understand the statement that superficial layers form "distinct" networks, since the authors did not record in deep layers, how do they know? It could be that the authors record from a brain area that is completely silent, including deep layers. Alternatively, it could be that only superficial layers are suppressed. Since it was shown previously that these experimental models of absence seizure have a focus, and since it is known that the focus is surrounded by a perifocal zone (the "penumbra"), where the activity is very much depressed (as in Schwatz & Bonhoeffer, 2001, for example). So it may be that the authors just record from this perifocal zone, and thus it would not be surprising to find a suppressed firing during the seizure.

The only way to further shed light into this would be to provide some recordings of deep layers in the same preparation. I understand that it may not be possible to image in deep layers, so I suggest to the authors to include a couple of unit recordings (simultaneous multi-unit recordings, from superficial and deep layers, would suffice). Only then, they can claim that the superficial network is distinct or disconnected.

If this is provided, it would greatly increase the value and impact of the study.

We thank the reviewer for this suggestion and have taken out the word “distinct” here. The reason we chose visual cortex was because our particular interest was in understanding how neocortical networks behave during the behavioral arrest induced by the absence seizure, not necessarily to probe the origin of the seizure focus. In agreement with the “penumbra” hypothesis outlined by the reviewer, we now have recordings from deep layers in visual cortex, which reveal similar findings regardless of depth of recording. These are shown explicitly in Figures 2-5.

2. Since layer 4 also seems suppressed during seizures, wouldn't this argue that indeed the whole area might be suppressed ? This also calls for having recordings in deep layers.

See above.

3. Similar findings were reported from single-unit recordings during focal seizures in human temporal cortex (Dehghani et al., Sci Reports 6: 23176, 2016, see the last figures in this paper), where it was also found that many neurons in Layer 2-3 are suppressed during the seizure. However, in that study, it was found that most of the suppressed neurons are RS (regular-spiking) cells, while those who increased firing are mostly FS (fast-spiking) cells, which were presumably inhibitory. This study should certainly be compared to the present results, and there should be a discussion of whether the ictal “low” and “high” cells found here could also be RS and FS cells, in which case it would completely agree with the phenomenon observed by Dehghani et al.

This would further stress the resemblance between absence seizures and focal epilepsy, which is another argument supporting a focal origin for absence epilepsy. I think this would be worth being discussed as well.

We have extended our discussion to compare our findings with the findings in focal-onset seizures as suggested and cited Dehghani et al. Although they found that presumed interneurons increased firing during focal-onset seizures in humans, we found that MGE-derived inhibitory neurons are largely suppressed during generalized absence seizures. We discuss how other interneuron subtypes may be involved, and the implications of these findings are now included in our discussion on page 11-12.

Reviewers' comments:

Reviewer #1 (Remarks to the Author):

I was pleased to see that the authors took the reviewer comments seriously and added considerable new experimental data and analysis to their paper. This is a case in which the resubmitted manuscript is really a significant improvement over the original. The authors addressed most of the points in my initial review (except point #1, see below). However, while the major issues have been resolved, I have a few remaining concerns I would like to see addressed, principally regarding reporting of statistics and methods:

Remaining issues:

1) I still have concerns about the statistical procedures used for significance testing, and I was sometimes unable to decipher the author's approach from the methods section.

For example, in Fig. 3A, how is the p-value being obtained for each 7 minute window? From lines 460-464 of the methods section, it appears that the distributions of all ictal/interictal time points within the window are being compared using a rank-sum test, but this is **not** acceptable since the time points are not independent (though it would certainly explain the egregiously small p-values).

A better method would be to sum the fluorescence/spiking within each ictal/interictal period and then compare the summed distributions. At the very least, the time points should be binned at a low enough rate that they are independent (and the adjacent-bin independence should be verified). The choice of analysis may affect the interpretation of whether neurons can "flip" from ictal-high to ictal-low, so it is important to get it right.

For the permutation tests, the authors should only use permutations that have lags >2 sec to avoid getting "redundant" permutations due to the slow course of calcium (or alternatively, use deconvolved data).

Most of the reported effects appear quite robust, so I'm not worried about the general strength and direction of the findings, but it is still incumbent on the authors to use the correct statistical reporting for all analyses.

2) I was not able to make sense of the autocorrelation analysis (lines 121-128). Please describe more clearly (perhaps include a supplementary figure showing the data) or remove from the paper if it is not necessary.

3) Readers should be able to reconstruct the experimental procedure and/or analysis from the methods section, and this is not always possible due to a general lack of detail (e.g, references to "custom code" with no further description). In particular:

Line 405: How were images registered? There are a number of widely-used methods

Line 415-416: Please explain how ROIs were detected in MATLAB

Line 432-433: Describe deconvolution approach

Line 491: Describe spike detection

4) There appears to be significant neuropil contamination in Fig. 1B, why not use neuropil subtraction throughout the paper? (lines 411-412 mention that this was done but the data was not shown). Note that this could affect pairwise CC values in Fig 4.

5) the authors use high laser power (up to 150mW) for imaging deeper layers. They mention that there was no bleaching by "visual inspection", but bleaching should be quantified. A common method is to measure the decrease in average fluorescence within frame to calculate the percent

decrease per minute (should be less than 1-2% decrease per min).

Despite the laundry list of concerns, I should emphasize that I am generally impressed with the revised manuscript and will support it's publication once these concerns are addressed.

Reviewer #2 (Remarks to the Author):

The authors addressed my major concerns and accordingly revised the manuscript. They removed overstatements and clarified the novelty of the paper which now focuses on the activity of the visual cortex in absence seizures rather than on contrasting the activity of layer 5 S1 cortex and layer 2/3 visual cortex which was the focus on the previous version.

Reviewer #3 (Remarks to the Author):

The authors have addressed my concerns and have improved the paper, I do not have further remarks.

March 30, 2018

Nature Communications Journal

Department of Neurology
One Baylor Plaza
Houston, Texas 77030-3498

Reviewer #1:

I was pleased to see that the authors took the reviewer comments seriously and added considerable new experimental data and analysis to their paper. This is a case in which the resubmitted manuscript is really a significant improvement over the original. The authors addressed most of the points in my initial review (except point #1, see below). However, while the major issues have been resolved, I have a few remaining concerns I would like to see addressed, principally regarding reporting of statistics and methods:

Remaining issues:

1) I still have concerns about the statistical procedures used for significance testing, and I was sometimes unable to decipher the author's approach from the methods section.

For example, in Fig. 3A, how is the p-value being obtained for each 7 minute window? From lines 460-464 of the methods section, it appears that the distributions of all ictal/interictal time points within the window are being compared using a rank-sum test, but this is **not** acceptable since the time points are not independent (though it would certainly explain the egregiously small p-values).

A better method would be to sum the fluorescence/spiking within each ictal/interictal period and then compare the summed distributions. At the very least, the time points should be binned at a low enough rate that they are independent (and the adjacent-bin independence should be verified). The choice of analysis may affect the interpretation of whether neurons can "flip" from ictal-high to ictal-low, so it is important to get it right.

We agree with the reviewer that it is important to have independent data points when comparing ictal and interictal time periods during the 7-minute windows using the ranksum test, so we adjusted the temporal analysis accordingly by concatenating all ictal frames and all interictal frames, and binning the resulting vectors in 2-second intervals for each ROI. The results are similar with *p*-values now within a more conventional range, and we have updated figure 3A, as well as all related results throughout the manuscript.

We have also verified that binning in 2-second intervals does result in independent data points. In 2 representative datasets, we first isolated the binned interictal and ictal activity inside non-overlapping 7-minute windows for all ROIs. We then computed the residuals of adjacent bins after subtracting the mean and then computed the autocorrelation function at lag 1 (i.e. for adjacent bins). Below we show the acf at lag 1 of the binned and original data from one representative neuron. For the binned data, the acf at lag 1 was <0.01, i.e. well within the 95%-confidence intervals (light blue lines, corrected for multiple comparisons between all cells in one dataset). Similar values were measured in all other

neurons. In the non-binned data, the acf at lag 1 did exceed the confidence interval, and this was the case for >90% of all neurons. Therefore, binning at 2-second intervals was effective in eliminating dependence between adjacent data points, as the reviewer suggested.

For the permutation tests, the authors should only use permutations that have lags >2 sec to avoid getting “redundant” permutations due to the slow course of calcium (or alternatively, use deconvolved data).

Thanks for raising this question. To a certain degree, we still think that shuffling the data by any random number of frames, like we did initially, makes sense statistically, because it takes into account all possible underlying activity dynamics that need to be separated out from the relationships with seizure times. On the other hand, the dependencies between data points separated by a few frames do disappear at intervals of 2 seconds, like we show above. It is reasonable to ask the question whether significance remains after the data get reanalyzed in this fashion to reassure the reader that calcium signal mediated dependencies between adjacent image frames do not significantly distort our result. Therefore, we re-analyzed all data for the participation analysis according to the reviewer’s request, using only permutations that were 2 seconds apart from each other when generating the null distributions. We updated figure 2, supplementary figure 5, and all related information in the text. Qualitatively, neither the null distribution nor the resulting graphs showing participation around seizure start and end times changed significantly.

Most of the reported effects appear quite robust, so I’m not worried about the general strength and direction of the findings, but it is still incumbent on the authors to use the correct statistical reporting for all analyses.

2) I was not able to make sense of the autocorrelation analysis (lines 121-128). Please describe more clearly (perhaps include a supplementary figure showing the data) or remove from the paper if it is not necessary.

We agree with the reviewer that this analysis does not provide information crucial to the main points of the paper, and have removed it from the manuscript.

3) Readers should be able to reconstruct the experimental procedure and/or analysis from the methods section, and this is not always possible due to a general lack of detail (e.g, references to “custom code” with no further description). In particular:

Line 405: How were images registered? There are a number of widely-used methods

Movies were motion-corrected along a 2D image plane (x-y motion). Motion parameters were estimated in the red channel, in animals where tdTomato-labeled interneurons were identified (green channel for all other recordings), by registering all image frames to the average of the first 5 image frames using a sub-pixel registration method (Guizar-Sicairos et al., 2008). Then, the correction parameters were applied to the green channel (in which calcium dynamics of cells were monitored) to reconstruct motion-corrected movies.

We added this information in the methods section and added this reference: Guizar-Sicairos, M., Thurman, S. T., and Fienup, J. R. (2008). Efficient subpixel image registration algorithms. *Opt. Lett.* 33, 156–158. doi: 10.1364/OL.33.000156.

Line 415-416: Please explain how ROIs were detected in MATLAB

The MATLAB routine was based on a previously published algorithm (Chen et al., 2013): A seed point for a local ROI was placed manually with a circular ring to cover the cell body. Around the center of the cell, the algorithm then computes an intensity profile in polar coordinates. The final selection of the pixels for the ROI matches the maximal intensity along the polar profile. This process was visually supervised for all ROIs to ensure the MATLAB routine did not select any pixels belonging to foreign cellular structures or neuropil.

This information was added to the methods section.

Line 432-433: Describe deconvolution approach

The method used to deconvolve the data we present in the text was based on a previously published method (Yaksi, Friedrich, 2006) that uses 1) an iterative smoothing process to remove local low-amplitude peaks representing noise without distorting the $\Delta F/F$ signal stemming from calcium fluctuations, and 2) inverse filtering of the smoothed traces with an exponential kernel. The other method was used for validation purposes and was also previously described (Vogelstein et al., 2010). It infers an approximation of the most likely spike train underlying the given fluorescence trace using an iterative expectation maximization algorithm.

We added this information to the methods section.

Line 491: Describe spike detection

APs were identified by searching for any time points t in the voltage trace that were characterized by a fast (0.25 ms) voltage increase, a peak lasting ~ 0.1 ms, and a fast return to baseline. In detail, these time points had to meet the following criteria: i) $V_m(t+0.25 \text{ ms}) > V_m(t) + x$, ii) $\text{mean}(V_m(t-\text{pre}_1 \text{ to } t-\text{pre}_2)) < \text{mean}(V_m(t+p_1 \text{ to } t+p_2)) - \alpha * x$, and iii) $\text{mean}(V_m(t+\text{post}_1 \text{ to } t+\text{post}_2)) < \text{mean}(V_m(t-\text{pre}_1 \text{ to } t-\text{pre}_2)) + \beta * x$, where the threshold x was 38 mV, pre_1 was 1.2 ms, pre_2 was 0.3 ms, p_1 was 0.25 ms, p_2 was 0.35 ms, post_1 was 2.5 ms, post_2 was 2.9 ms, α was 0.5 and β was 0.8 for whole-cell recordings, and x was 1.9 mV, pre_1 was 0.8 ms, pre_2 was 0.25 ms, p_1 was 0.15 ms, p_2 was 0.24 ms, post_1 was 1.8 ms, post_2 was 2.1 ms, α was 0.55 and β was 0.45 for cell-attached recordings. Every voltage trace used in the analysis was visually inspected to ensure no false positive or negative spike detections, and only traces that were stable over at least 10 seizures were used.

We added this information to the methods section.

4) There appears to be significant neuropil contamination in Fig. 1B, why not use neuropil subtraction throughout the paper? (lines 411-412 mention that this was done but the data was not shown). Note that this could affect pairwise CC values in Fig 4.

We agree that this is an important technical issue. However, figure 1B does not show significant neuropil contamination, given several instances of a number of cells being inactive while there is ongoing strong neuropil activity. If the neuropil was optically contaminating all cellular signals significantly, this should not happen. Please note that neuropil contamination is a more prominent issue in acute, bulk-loaded calcium indicator recordings where all extracellular structures take up the dye and neuropil signal amplitude is on par with cellular fluorescence, than with virally expressed genetic indicators, where neuropil signals are typically much weaker compared to cellular signals. This is the case in our recordings, as one can see in figure 1B: The maximum amplitude of the DF/F signal of neurons, such as the ictal low example (blue trace) typically reached $\sim 350\%$, whereas the neuropil signal usually did not exceed $\sim 200\%$.

Having said that, for a greater degree of certainty, we did re-analyze 2 representative datasets, one from layer 2 and one from layer 5. First, we computed the level of neuropil contamination present in these recordings by measuring the ratio of the signal inside neuron-sized blood vessels and the surrounding neuropil (this is according to a method we adopted in Lee et al., 2017). Here, this ratio was, on average, 0.5. The layer 5 dataset originally had 67% ictal low neurons and 14% ictal high neurons, and the layer 2 dataset had 95% ictal low neurons and 2% ictal high neurons. After neuropil correction, these numbers changed to 60% ictal low and 16% ictal high in the Layer 5 dataset, and 85% ictal low and 4% ictal high neurons in the layer 2 dataset. This is as expected: since the neuropil is overwhelmingly ictal low subtracting it reduced the number of ictal low cells and increased the number of ictal high cells slightly. In addition, mean pairwise correlation coefficients dropped from interictal to ictal time periods by 0.03 (L5) and 0.09 (L2) before, and by 0.03 (L5) and 0.07 (L2) after neuropil correction. These observations do not change the basic conclusions of this study.

Please note that the above procedure also carries a risk of subtracting a real signal that is common between the neuropil and the cells. For example, there are many instances where neuropil activity coincides with the activity of a number of cells, not because of optical neuropil contamination but because inputs are shared, as well as some of the processes in the adjacent neuropil patch may come from the cell itself or its neighbors, with which the cell's activity may be correlated. Therefore, completely subtracting from cell activity the component that is parallel to adjacent neuropil activity, would also eliminate a part of the cell's own activity, introducing significant error. This is why the coefficient of the subtraction (optical contamination) has to be estimated from the experimental sample each time (see Lee et al., 2017, and Kerlin et al., *Neuron*, 2010). Since the estimate of the subtraction coefficient is typically an average estimate that applies across the cells in the FOV, it is possible that the subtraction itself can introduce error.

In summary, we confirmed that doing the neuropil correction does not alter the basic conclusions of the paper, even though it leads to slightly lower estimates of ictal low cells and slightly higher estimates of ictal high cells. However, we chose not to alter the figures in the paper since, as explained above, the neuropil correction itself is not error free and would run the risk of underestimating the true number of ictal low cells, which are the main focus of this paper.

5) the authors use high laser power (up to 150mW) for imaging deeper layers. They mention that there was no bleaching by "visual inspection", but bleaching should be quantified. A common method is to measure the decrease in average fluorescence within frame to calculate the percent decrease per minute (should be less than 1-2% decrease per min).

We did proceed to analyze the average fluorescence over time in all 9 layer 5+6 recordings. When averaged across those datasets, the fluorescence dropped to 95% (+/-1.7 sem) after 10 minutes, and to 93% (+/3.9 sem) after 25 minutes, which means it reduced by less than 0.5% per minute. The recording with the largest drop in fluorescence only went down to 79% after 25 min (0.84%/min). We are therefore very confident that we did indeed cause minimal bleaching at the laser power values we used.

Sincerely,

Jochen Meyer, PhD
Instructor of Neurology
Baylor College of Medicine

Jeffrey L. Noebels, MD, PhD
Professor of Neurology,
Neuroscience, and
Molecular Genetics
Baylor College of Medicine

Atul Maheshwari, MD
Assistant Professor of
Neurology
Baylor College of Medicine

Stelios Smirnakis, MD, PhD
Department of Neurology
Brigham and Women's
Hospital
Harvard Medical School

REVIEWERS' COMMENTS:

Reviewer #1 (Remarks to the Author):

The authors addressed all of my concerns.